# Current Functionality and Potential Improvements of Non-Alcoholic Fermented Cereal Beverages

**DOI:** 10.3390/foods9081031

**Published:** 2020-08-01

**Authors:** Maria Valentina Ignat, Liana Claudia Salanță, Oana Lelia Pop, Carmen Rodica Pop, Maria Tofană, Elena Mudura, Teodora Emilia Coldea, Andrei Borșa, Antonella Pasqualone

**Affiliations:** 1Department of Food Engineering, Faculty of Food Science and Technology, University of Agricultural Sciences and Veterinary Medicine Cluj-Napoca, 400372 Cluj-Napoca, Romania; maria.socaci@usamvcluj.ro (M.V.I.); elena.mudura@usamvcluj.ro (E.M.); teodora.coldea@usamvcluj.ro (T.E.C.); andrei.borsa@usamvcluj.ro (A.B.); 2Department of Food Science, Faculty of Food Science and Technology, University of Agricultural Sciences and Veterinary Medicine Cluj-Napoca, 400372 Cluj-Napoca, Romania; oana.pop@usamvcluj.ro (O.L.P.); carmen-rodica.pop@usamvcluj.ro (C.R.P.); maria.tofana@usamvcluj.ro (M.T.); 3Department of Soil, Plant and Food Sciences, University of Bari ‘Aldo Moro’, Via Amendola, 165/A, 70126 Bari, Italy; antonella.pasqualone@uniba.it

**Keywords:** cereal beverage, fermentation, functional, non-alcoholic, health benefits

## Abstract

Fermentation continues to be the most common biotechnological tool to be used in cereal-based beverages, as it is relatively simple and economical. Fermented beverages hold a long tradition and have become known for their sensory and health-promoting attributes. Considering the attractive sensory traits and due to increased consumer awareness of the importance of healthy nutrition, the market for functional, natural, and non-alcoholic beverages is steadily increasing all over the world. This paper outlines the current achievements and technological development employed to enhance the qualitative and nutritional status of non-alcoholic fermented cereal beverages (NFCBs). Following an in-depth review of various scientific publications, current production methods are discussed as having the potential to enhance the functional properties of NFCBs and their safety, as a promising approach to help consumers in their efforts to improve their nutrition and health status. Moreover, key aspects concerning production techniques, fermentation methods, and the nutritional value of NFCBs are highlighted, together with their potential health benefits and current consumption trends. Further research efforts are required in the segment of traditional fermented cereal beverages to identify new potentially probiotic microorganisms and starter cultures, novel ingredients as fermentation substrates, and to finally elucidate the contributions of microorganisms and enzymes in the fermentation process.

## 1. Introduction

Consumers’ lifestyles have changed in recent years and will continue to change by being influenced by globalization, economic growth, rapid advances in food science and technology, and/or lifestyle choices and religious restrictions [1,2,3]. With regard to consumption behaviour, sensory acceptance is still the main choice criteria for consumers [4,5,6] and it is strongly dependent on cultural backgrounds, as well as previous sensory exposure to a specific food product [7,8].

Beverages are an optimum vehicle to transport nutrients and bioactive compounds into the body as well as to facilitate their bioavailability. Bioactive compounds, such as phytochemicals (e.g., phytoestrogens, phenolic compounds, flavonoids, carotenoids, etc.), dietary fibre, vitamins, fatty acids, probiotics, and minerals, can be incorporated into beverages. The presence of these compounds offers the prospect of using food as a valuable element in disease prevention strategies, particularly in the early stages of the diseases [9,10,11]. Worldwide, statistics clearly show the growing trend of functional beverage consumption [12], due to their nutrient contents, convenient packaging, design, ease of transportation and storage, and for their shelf-stable nature [13].

Functional beverages could be classified as dairy based, fruit and vegetable based, legume based, cereal based, coffee, or tea. The functionality traits of these beverages address different needs and lifestyles: to boost energy, to fight the ageing process, fatigue, and stress, or to target diseases [14].

Fermentation is widely used to improve the nutritional value, the digestibility level, shelf life, functional properties, texture, taste, and flavour of the beverages [15,16,17,18]. A popular class of fermented beverages are those made from cereals: barley (*Hordeum vulgare* L.), maize (*Zea Mays* L.), millet (*Panicum miliaceum* L.), oats (*Avena sativa* L.), rice (*Oryza glaberrima*/*Oryza sativa*), rye (*Secale cereale*), sorghum (*Sorghum bicolor*), and wheat (*Triticum aestivum* L.) [19]. These cereals are a good fermentation substrate and also pose a potentially functional trait, as they contain nutrients that can be easily assimilated by probiotics [20]. The processing steps could include soaking, sprouting, malting, cooking, grinding, and filtering. The fermentation of cereals is influenced by several factors (e.g., length of fermentation, temperature and pH, moisture content of grain, growth factor requirements, cereal nutrients, etc.) which require control by using technological methods to standardize quality.

Fermented beverages are affordable, and their production involves traditional methods to maintain hygiene conditions, product quality, and security [19,21]. Lactic acid bacteria (LAB) dominate the fermentation process and lead to a low pH, which is incompatible with the development of pathogenic bacteria, thereby increasing the shelf life and product safety [22]. Traditional methods exploit mixed cultures of various potentially beneficial microorganisms, referred to as probiotics [23]. Fermented beverages are considered essential because they serve as vehicles for beneficial microorganisms that play an important role in human health, and also for their nutritional, nutraceutical, and pharmaceutical properties [21,22,23,24]. Moreover, it was shown that fermentation improves protein digestibility and the bioavailability of minerals and other micronutrients [25].

The consumption of these beverages has the potential to reduce the adverse health and economic impacts of poor diets. Additionally, among the main benefits of cereal-based beverages, there is the possibility of consumption by vegetarians, vegans, and lactose-intolerant consumers [26]. This paper outlines the current achievements and further development for enhancing the functionality of the non-alcoholic fermented cereal beverage (NFCB) segment.

## 2. Non-Alcoholic Fermented Cereal Beverage Segment

### 2.1. An Overview of NFCBs

Reviewing the existent literature regarding some of the most studied NFCBs, it is shown that they are usually based on barley, maize, millet, oats, rice, rye, sorghum, and wheat [27,28]. Cereals are known as important sources of dietary proteins [29], energy, carbohydrates, vitamins, minerals, and fibre (arabinoxylan and *β*–glucan), but, at the same time, they are deficient in some basic components (essential amino acids, e.g., lysine) [19,30]. Whole grains are known to have important bioactive compounds and high nutritional values and their regular consumption has a positive effect on health [31,32,33]. The presence of soluble fibre lowers the glycaemic index of the beverage by slowing down its digestion and absorption, whereas phenolic compounds have antioxidant potential and scavenge harmful free radicals in the body [9].

Most of the traditional and currently produced NFCBs are considered functional foods and wholesome nutritional products [22,34,35]. Functional foods have been intensively studied in recent years and are continuously looked at as research efforts turn to processing food matrices, such as cereals, vegetables, and fruits, into medicine-like products.

A more detailed definition describes functional foods as industrially processed or natural foods that, when regularly consumed within a diverse diet at efficacious levels, have potentially positive effects on health outside basic nutrition [36]. This means that such products should provide a therapeutic benefit if consumed regularly within a diverse diet and with the condition of having the main nutrients extracted in a standardized manner and dosage [37].

Originally, cereal fermented beverages were mainly produced due to the need for conserving and utilizing various cereals and crops with affordable financial implications. Some of them, which are now commercially available as soft drinks and non-alcoholic beverages, were traditionally prepared as alcoholic beverages, having higher contents of alcohol [38].

Currently, “non-alcoholic” is a regulatory term and the laws regarding it vary across the globe. For example, EU regulation no. 1169/2011 simply states that a beverage must be labelled as an alcoholic drink if it has an alcoholic strength by volume (ABV) of over 1.2%. Moreover, in Great Britain, an alcohol-free drink has a maximum alcohol content of 0.05 % ABV; in Germany, a beverage is considered alcohol-free if the maximum alcohol limit is below 0.5 % ABV; in Spain, a non-alcoholic beer contains a maximum of 1 % ABV; in France, alcohol-free beers contain a maximum of 1.2 %; in US, non-alcoholic beers are of a maximum 0.5 % ABV; in China, non-alcoholic drinks can be of up to 0.5 % ABV [39]; in Japan, non-alcoholic beverages contain up to 1% ABV [40]. Depending on the country’s traditional recipe, *boza*, a lactic acid fermented drink, has an alcohol content of less than 1% ABV in Turkey and up to 7% ABV in Egypt [41], probably due to microorganisms involved in the fermentation process [42].

Fermented drinks were and are obtained using a combination of fermentable substrates, like cereal mixes, fruits, plants, spices, legumes, and vegetables. The possibility of combining various fermentable substrates and of adding supplementary bioactive compounds to the final product encourages the development of different versions of the same original recipes, according to specific resources, health concerns, and nutritional needs. Such enhancements can be seen, for example, in vitamin A-fortified *mahewu* [43] or in establishing the suitable temperature for saccharification and oligosaccharide production efficiency in *amazake*, containing up to 0.5 % ABV [44].

Globally, there are numerous similar non-alcoholic cereal fermented beverages with similar names and profiles targeted for thirst quenching properties, on nutrition added value, cultural significance, and on providing alternatives to alcoholic beverages. Fermented beverages based on cereals are somewhat common around the world as staple foods, particularly in developing countries and are all made in a similar manner, generally using spontaneous microbial cultures [38]. Although there are various non-alcoholic cereal fermented beverages with different chemical profiles and sensory traits, all of them present certain bioactive compounds and therapeutic agents, which make them beneficial for human health. The functional compounds available and detected in NFCBs have variable values, therefore, it is difficult to assess their functional impact when consumed regularly [17]. If cereals lack certain nutrients, then additional food matrices can be added to enhance the final NFCB, as seen in a case of *mahewu* enhanced with *Moringa oleifera* leaf powder for elevating Ca and Fe contents [45]. Another example of a functional NFCB is a non-alcoholic beverage made of green tea and barley malt wort for delivering superior amino acid content [46].

In Table 1, there are several NFCBs listed with their place of origin, constituent cereals, microorganisms, nutritional compounds, and potential health benefits. As previously mentioned by other authors, the functional outcome of such products is strongly connected to the selection of the cereals, microorganisms, fermentation temperature and time, and other additional food matrices [47,48,49].

Some of the NFCBs presented in Table 1 are commercially available, with various scales of production, while others are mainly homemade beverages, produced for individual consumption using traditional methods. *Busa*, *kunun-zaki*, *mahewu*, *munkoyo*, *obushera*, *oshikundu*, *pozol*, and *tobwa* are produced in rural and urban areas and their production is essentially a home-based industry as for now, there is no large-scale factory production. *Amazake* is commercially available in Japan and it is classified as a soft drink [90,91]. *Borş* is also industrially produced and used as a flavour enhancer in Romanian gastronomy or it is consumed as a refreshing drink [55]. *Boza* is one of the most popular Bulgarian beverages and is industrially produced at a large scale in all countries of the Balkan Peninsula [92]. *Kvass* is a traditional fermented Slavic and Baltic beverage and is one of the most popular beverages in Russia, numerous varieties emerged due to its popularity and market demand. Currently, *kvass* production is designed according to traditional processes, with the implementation of modern biotechnological methods [72]. *Shalgam* is a traditional beverage of southern Turkish cities and it is commercialized in various markets of European cities [86]. In Table 2, there are more details presented regarding NFCBs and their sensory properties, pH values, nature of use, fermentation status, and production scale.

### 2.2. Processing Technologies and Their Outcome

The dietary attributes and sensory traits of cereal products can be at times viewed as inferior or deficient in comparison with those of milk and milk-based foods. Some of the reasons behind this include the smaller protein quantities, deficiency in certain amino acids (lysine), the presence of antinutrients (phytic acid, tannins, and polyphenols), and the coarse nature of grains [19].

The fermentation of starchy sources is more complex compared to that of low molecular sugars (glucose or sucrose) because, in general, the starch must at first be converted into fermentable sugars. To achieve an almost complete starch degradation, two main types of amylolytic enzymes are required (α-amylase and glucoamylase) [98].

Several methods have been engaged with the aim of enhancing the nutritional qualities of cereals. These include genetic improvement and amino acid supplementation with protein concentrates or other protein-rich sources, such as grain legumes or defatted oil seed meals of cereals. Additionally, several processing technologies, which include cooking, sprouting, milling, and fermentation, have been put into practice to improve the nutritional properties of cereals, although the best one is probably fermentation. In general, the spontaneous fermentation of cereals leads to a decrease in the level of carbohydrates, as well as some non-digestible poly- and oligosaccharides [35]. Certain amino acids may be synthesized, and the availability of B group vitamins may be improved [99]. Fermentation also provides optimum pH conditions for the enzymatic degradation of phytate, which is present in cereals in the form of complexes with polyvalent cations, such as iron, zinc, calcium, magnesium, and proteins. Such a reduction in phytate may result in a several fold increase in the amounts of soluble Fe, Zn, and Ca [19,100,101,102].

The current article advances one general processing technique, by compiling all the traditional recipes assessed and integrating germinated and non-germinated grains, to adapt it to an industrial scale with functional improvement. Several traditional processing steps can be applied in the production. The general outline of the process is essentially the same and that presented in Figure 1: the grains, after conditioning, are either soaked/wet milled or the grains are dry milled and the flour is extracted in water afterwards, the mix is boiled to gelatinize the starch, a source of enzymes to hydrolyse the gelatinized starch into fermentable sugars is added, and finally, spontaneous fermentation occurs [19,103].

#### 2.2.1. Pre-Treatment of Raw Materials

Cereal processing is an essential component of the brewing production chain and the milling process is the main procedure. Before milling, the cleaning and conditioning of the grains is required. The cleaning process allows for the removal of various impurities, depending on the raw material types. The most common grain impurities are shrivelled grains, other cereals, grains damaged by pests, grains with discoloured germs, and sprouted grains. There are also miscellaneous impurities, including extraneous seeds, damaged grains, extraneous matter, husks, ergots, decayed grains, and insects. The conditioning or tempering of grains is performed using the monitored addition of water, which turns the endosperm softer and the bran harder. Doing this prevents the bran from breaking up, aids gradual separation throughout milling, and enhances sieving efficiency [104].

There are two milling categories, namely dry and wet milling, each having its own characteristics. Dry milling removes the germ and the outer fibrous materials of grains, as these by-products are not used in traditional ways [105]. Malting utilizes the power of natural germination when the grains, after absorbing water, germinate in the presence of oxygen to achieve a moisture content of up to 47% [106]. Another by-product is represented by floating kernels, which are unsuitable for malting. Throughout germination, the grain’s embryo expands, and rooting begins. Moreover, the germination and steeping processes frequently overlap. It is recommended to keep the germination time and temperature at low values, given that long and warm germination processes lead to longer roots, resulting in larger malt yield losses [104,106].

During germination, grain enzymes start to break down the endosperm high-molecular-weight material into easily digestible components for the yeasts. Drying the malt stops the germination process [104].

In the case of traditional methods, the grains are usually superficially cleaned. A part of the cleaned grains (in variable percentages) is washed and soaked (10 to 20 h) at room temperature (26–35 °C). The soaked grains are drained and left for germination for 48 to 72 h with a frequent spraying of tap water. The germinated grains are sun dried (5 to 20 h), so the success of the process depends on the weather, mostly the intensity of the sunshine. Afterwards, the grains (malted and non-malted) are milled separately or in a mixture to obtain flour using rudimentary equipment [107].

Industrial cleaning processes aim at removing impurities and all other materials except for grains, using specific equipment such as magnetic separators, disc or sieve separators, aspirators, destoners, colour sorters, etc. The conditioning process ensures the complete hydration of grains, holding them in suitable containers for specific time intervals. Usually, depending on the grain varieties and initial moisture levels, the soaking time and temperature may be different [104].

For industrial-scale malting (Figure 2), the cereals are dried, and kilning is used to stop further transformations.

During controlled drying, the water content should go under 5%, to stop the enzymatic activity whilst colour and flavour compounds are formed. A lot of by-products can additionally be recovered and further capitalized at an industrial scale. Dry milling can be extended to pearling, an abrasive procedure for gradually removing the testa and pericarp, aleurone and sub aleurone layers, and the germ. This results in polished grains and by-products with enhanced contents of bioactive compounds. Alternatively, wet milling, mostly used for producing starch and gluten, can increase the value of the cereals, as it is a source of by-products, with coproducts such as steep solids (abundant in compounds of pharmaceutical interest), germs (used in the oilseed-crushing industry), and bran [104].

#### 2.2.2. Mashing

The traditional mashing process differs greatly, depending on the local culture, and has low efficiency. In Africa, either sorghum or finger millet malt is commonly used as a source of enzymes [22,108]. In the case of gowé production, a beverage based on maize and sorghum, the grains are milled to flour, which is optionally mixed/kneaded with tap or hot water or with a supernatant of a previous production and left to ferment at room temperature. Additionally, this processing technique can include a saccharification step, where a part of the malted sorghum or maize flour is kneaded with water and left for saccharification for 5 to 10 h [107]. For munkoyo, a maize-based beverage with a variable ABV, the specific feature is the usage of *Rhynchosia* roots or watery root extract as an enzymes source [22,77,78]. Starch gelatinization facilitates the activity of α- and β-amylases from the *Rhynchosia* roots for at least 4 h up to a maximum of 24 h, which hydrolyse the starch into fermentable sugars [22,103,109]. Mahewu is an example of a non-alcoholic sour beverage made from corn meal, consumed in Africa and some Arabian Gulf countries [74]. It is prepared from maize porridge further mixed with water. Sorghum, millet malt, or wheat flour is then added and left to ferment [19]. The production techniques of obtaining kvass, a cereal-based beverage produced from rye and barley malt, rye flour, and stale rye bread, uses as raw material either stale sourdough bread or malt rye malt and rye flour [110]. Boza is a colloid suspension, non-alcoholic beverage consumed in Bulgaria, Albania, Turkey, and Romania, made from wheat, rye, millet, maize, and other cereals mixed with sugar or saccharine [92]. Boza’s preparation involves six stages: the preparation of raw materials, boiling, cooling, straining, the addition of sugar, and fermentation. Another option for its production is the use of previously fermented boza as an inoculum [59].

The industrial mashing process resembles the beer production process. Between one and three volumes of water are added per volume of milled grains, and during the cooking process, the mixture turns into a mash. The mixture is cooked at a normal pressure or in an autoclave for about 2 h at 4–5 atmospheres.

The macromolecular profile of cereal-based beverages is generally determined by polymeric compounds (proteins, polysaccharides, and polyphenols) and their progress in depolymerization during processing [111,112]. Given that yeasts or specific strains of lactic acid bacteria (LAB) cannot metabolize high-molar-mass substances [113], the macromolecules are solely depolymerized during the malting and mashing process by the malt’s intrinsic enzymes. During malting, the enzymatic degradation of polymers is technologically controlled by the degree of steeping, germination time, and germination temperature [114,115]. Modern brewing barley varieties are bred to be balanced in malting performance and to meet the required brewing specifications. The degradation of starch into fermentable sugars (amylolysis) is the primary objective of mashing (substrate production for fermentation) [116], since during the subsequent fermentation process, only low-molar-mass compounds, fermentable sugars, and low-molar-mass proteinaceous compounds are metabolized by microorganisms (e.g., LAB and yeasts) [117,118].

Lautering is the next step in the large-scale process, through which solid and liquid fractions, respectively, the spent grain, composed of sugar-extracted grist or solids remaining in the mash, and the sweet wort with high contents of fermentable sugars are obtained. The spent grain is the major by-product of the brewing industry and represents a valuable source of bioactive ingredients and a potential ingredient for functional foods [119]. In small-scale production, malt extract can be used instead, thus skipping the use of grain malt (including milling, mashing, and lautering) [120].

#### 2.2.3. Cooling and Addition of Yeast, LAB Cultures, and Other Ingredients

After boiling, the mash is gradually mixed with cold water in a 1:1 ratio. This slurry is often filtered or decanted to remove the grinding waste and insoluble plant material. In many traditional processes, where cereals are soaked in water for a few days, a succession of naturally occurring microorganisms will result in a population dominated by LAB. In such types of fermentations of endogenous grains, amylases generate fermentable sugars, which serve as energy sources for lactic acid bacteria. When no malted cereals are used, sucrose is added to the beverage to mimic the malt’s sweet taste.

The bacterial flora formed in each fermented cereal drink is influenced by several factors, such as water activity, pH level, salt concentration, temperature, and the composition of the grain matrix, which must be considered in industrial processes. However, most fermented drinks, including the well-known products commonly met in the Western world, as well as those beverages of other origins which are less well studied and characterised, rely on lactic acid bacteria to mediate the fermentation process [19,121]. Lactic acid fermentation contributes towards the nutritional value, shelf life, safety status, and acceptability of a wide range of cereal-based foods [118]. Fermentation is often just one step in the process of fermented food preparation. Other operations, such as volume reduction, salting, or heating, also affect the final product characteristics [19,122]. Depending on the desired product, further steps can be applied afterwards, such as standardisations and the addition of other ingredients like flavourings, sugar, and stabilizers.

#### 2.2.4. Fermentation Process

Despite the lack of process control, dealing with unstandardised microbial flora composition, delayed fermentation, and imperfect reproducibility of the fermentation process, spontaneous fermentation offers complex microbial diversity, providing higher levels of intrinsic stability to the microbial community [123]. This is achievable due to stabilizing interactions between species that prevent and inhibit the proliferation of unwanted microorganisms, including pathogenic ones. Recently, the existence of stability criteria for complex microbial communities was proven [124]. In terms of spontaneous fermentation, this can be explained by the resilience to small perturbations when there is a balance between the availability of resources, namely nutrients, and consumers (e.g., lactic acid bacteria). Commonly, the production of artisanal fermented beverages is conducted in successive batches [55], by using a natural starter from the previous fermentation. At all traditional sites, spontaneous fermentation proceeds after cooling. A 24 h fermentation period is sufficient for some traditional beverages to develop their characteristic sensory attributes, although in practice, fermentation time can go on for up to three days. Some brews, obtained through a 6 to 15 h fermentation interval of non-malted grain flour, are enhanced with commercial sugar (sucrose) to obtain the sweet taste. This process is a distortion of the original germination technique for gowé production [22]. A modern industrial process is different from the traditional process regarding the introduction and control of thermophilic LAB cultures, which only produce lactic acid, the extension of the product’s shelf life by pasteurization and/or chemical preservation, and the inclusion of sugar and/or artificial sweeteners [125,126]. Controlled fermentation also leads to a general improvement in the shelf life, texture, taste, and aroma of the final product. During cereal fermentation, several volatile compounds are formed, which contribute to a complex blend of flavours [127]. Moreover, there is a good opportunity to apply colloidal dispersions in the form of nanoemulsions to deliver food grade nanoparticles, which contain water-insoluble molecules that were formerly unsuitable due to their poor soluble characteristics. Thus, there is a wide range of healthy foods which can be further designed, such as cereal-based fermented beverages enhanced with nanomolecules possessing beneficial health attributes [27].

### 2.3. Fermentation Microbiota and Safety of NFCBs

Originally, fermented beverages were only consumed in their native regions, however, due to increasing demand and interest, some of those traditional beverages are available to international markets. The attributes of traditional fermented beverages are influenced by several factors, such as the use of different raw materials, manufacturing methods, natural microbiota, and fermentation conditions. The microbiology of many traditional fermented drinks prepared from the most common types of cereals is quite complex.

There is a lot of diversity in the traditional processing techniques used for cereal-based fermented beverages all over the world, integrating single or multigrain cereals, germinated and/or non-germinated grains. Many types of cereal-based fermented beverages are produced in Africa, such as *togwa* in Tanzania, *mahewu* in Zimbabwe and South Africa, *maw*è in Benin, and *munkoyo* in Zambia and the Democratic Republic of the Congo [9,19,128,129]. Generally, the preparation of these products is a traditional family activity with an uncontrolled fermentation process by diverse microbial communities. The composition of the microbial community in a fermented food product largely determines the key product properties [130,131]. In other words, variations in microbial communities may result in differences in product quality, taste, acceptability, and microbial stability.

In one of our previous papers [55], we proved the impact of processing parameters, namely temperature and batch fermentation cycles, on the chemical composition of borș. Interestingly, the final composition of cereal-based beverages might not only be influenced by the physical process parameters. Processing practice variation affects the microbial composition of the fermenting microbial community. Despite the decreased pH caused by the lactic acid bacteria activity, the low pH of munkoyo also permitted the development of acidifying bacteria [22]. An important role is played by the initial microbial composition of the raw materials used in the process. The same study also referred to the fact that further investigations might be needed into the soil composition of the harvested raw materials. Along with the prevention of growth of most pathogenic strains [132], a low pH (of 2.5 to 3.5) improves food safety and expands the shelf life of this type of beverage [22]. The pathogenic microorganisms that are aerobes and facultative anaerobes and ferment simple sugars have an optimum pH for growth of 6.0 to 8.0. However, growth can occur at a pH as low as 4.3 and as high as 9, but with a combination of factors (pH, water activity), the control of foodborne pathogen growth can be done [133].

The fermentation of most cereals is natural and involves mixed cultures of yeasts, bacteria, and fungi [134]. Some microorganisms may participate in parallel, while others act in a sequential manner with a changing dominant flora during fermentation. The challenge, though, is the generally uncontrolled nature of the fermentation, which raises safety concerns, as well as the lack of standardization in the methods used, thus further research and development are needed to improve the traditional fermentation processes [80]. In this regard, the introduction of starter culture technology has led to greater consistency and safety and to better product quality [135].

Furthermore, the main difficulties encountered in the production of cereal-based beverages using traditional processes are linked to the high variability of unit operations and the unhygienic conditions of the processing environment. The soaking and germination parameters (temperature, duration, moisture) vary within and between processors. Moreover, during soaking and germination, the grains can be infested by fungi with the potential development of mycotoxins (aflatoxins) [136].

The continuous study of the fungi, yeasts, and bacteria strongly involved in ensuring a certain quality of the NFCB allows for the optimization of same-product delivery [85,137,138]. Such upgraded fermented beverages are sometimes the outcome of general efforts to enhance original recipes [139]. *Kunun-zaki* is an example of a traditional wheat and sorghum fermented beverage now also commercially available in the form of a powder [68], and there are several strategies proposed to upgrade and re-engineer the process of *gowé* production, a beverage obtained from sorghum and maize [66]. As seen so far, it is possible to prolong shelf life through the co-incubation of probiotic cultures, as seen in the development of some cereal-based fermented beverages [37,140] or to market to a group of consumers looking for healthy and functional foods by using oats, for example, as a main ingredient [141]. In-depth studies are still being conducted to ensure food safety in the processing technology of fermented foods through the keen selection of starter cultures and thorough examination of the specific microorganisms [142,143]. For example, it was recently shown that *Enterococcus faecium* YT52 isolated from *boza* is susceptible to clinically relevant antibiotics and contains low numbers of virulence factors and antibiotic resistance genes. Therefore, the enterocin-producing *E. faecium* YT52 strain poses a low risk to consumer health, and this strain may be used as a starter or a co-starter culture for improving the food safety of fermented products by acting against foodborne pathogens, such as *Listeria monocytogenes* and *Bacillus cereus* [144]. Moreover, rethinking the technological processes for obtaining various cereal-based fermented beverages helps to increase their functionality and overall therapeutic and nutritional properties. Such an example is *boza*, enhanced by fermenting cereals with *Lactobacillus acidophilus*, *Bifidobacterium bifidum*, and *Saccharomyces boulardii*, and by adding chickpea flour to the fermentable substrate for an elevated protein content [145]. Along with these interventions, narrowing the risks of product spoilage through the inoculation of specific strains of LAB and yeasts and processing the cereals in a certain manner repetitively allows the producers to deliver same-quality functional NFCBs.

Fermentation is one of the oldest known biotechnologies and the most economical procedure for producing new foods and ensuring product conservation. The yeasts responsible for fermentation in fermented drinks include species of *Saccharomyces*, *Saccharomycopsis*, *Schizosaccharomyces*, *Pichia*, *Candida*, *Torulopsis*, and *Zygosaccharomyces*. Brewer’s yeast, *Saccharomyces cerevisiae,* metabolizes various sugars, mainly into alcohol, but also into other flavour-active substances. The most used microorganisms in fermentation processes belong to *Lactobacillus* species, which synthesize many flavour-active substances and lactic acid and are considered probiotic microorganisms known to support intestinal microbiota [146]. Fermented beverages are known to be rich in bioactive compounds, such as immune globulin peptides and the bioactive hormone cytokinin [19,71,82,147,148]. The use of a wide variety of microorganisms and yeasts is implied in the production of NFCBs by both traditional and modern means.

Largely responsible for the fermentation process are the indigenous microbiota present on the substrate or which can be added artificially as a culture. Basically, there are four main fermentation processes, which include alcohol production, lactic acid development, acetic acid production, and alkaline fermentation [19]. The modern industry for fermentative foods and beverages is innovative, given that it currently employs thermophilic fermentation, DNA technologies, molecular devices, designed starter cultures, genetic engineering, etc. Recombinant DNA technology has provided new insights for enhancing the product quality by designing tailor-made starter cultures that perform better than those found naturally [60]. For example, Basinskiene et al. used *Lactobacillus sakei* KTU05-6 and *Pediococcus pentosaceus* KTU05-10 to ferment extruded rye for obtaining *kvass*, a traditional Lithuanian NFCB. They showed that the pH of the beverages fermented by LAB reached lower values compared to yeast fermentation and a they had a higher amount of organic acids. Innovative technology was applied, such as xylanolytic enzymes and antimicrobial active LAB, to improve the product’s functional properties [71].

Functional cereal fermented beverages are becoming more attractive as they represent healthy alternatives for lactose intolerant consumers and for those who avoid certain allergens. For example, a study describes the production of kefir-like riboflavin-enriched beverages based on oat, maize, and barley flours. To obtain this beverage, riboflavin-producing Andean LAB were used, consisting of five *Lactobacillus plantarum* strains and two *Leuconostoc mesenteroides* strains [149].

As previously discussed, ensuring product safety, although difficult at times, is a crucial step in the production of traditional beverages where the fermentation process is spontaneous. Thus, microorganisms found on the brewers’ skin, hair, and clothes can alter the product safety, therefore, high standards of hygiene are mandatory. Nevertheless, through the identification of bacteria strains and yeasts, the safety and quality status of the fermented beverages are guaranteed. A safety case study was conducted concerning “*obushera*”, a Ugandan traditional fermented cereal beverage where important steps, such as pasteurization and ensuring water quality, are in the loop to ensure product safety and quality, as pathogens can also change the product’s sensory characteristics [80].

Superior results can be obtained in the production of NFCBs by better understanding the interaction of microorganisms in the fermented substrates. The beneficial outcomes of controlling microorganisms are mirrored in product safety, increased shelf life, improved nutrient contents and availability, palatability, and enhanced sensory traits. For example, it was shown that *S. cerevisiae* improves LAB growth by transmitting essential metabolites, such as pyruvate, amino acids, and vitamins, while it uses some metabolites produced by LAB as carbon sources [150]. Moreover, Salari et al. concluded in their study that malt and *L. delbrueckii* were the best substrate and lactic strain for producing a functional beverage with the highest cell viability (1.2 × 10^6^ cfu/mL after 4 weeks) [151].

#### Probiotics

Non-alcoholic fermented cereal-based beverages contain a wide range of diverse probiotics, depending on their cereal substrate and overall production methods. The processing method should assure the stability of the bacterial composition in order for the final product to possess probiotic functionality [152]. Still, the threshold to declare a beverage a probiotic one must be higher than 10^7^ CFU/mL. Moreover, not all lactic acid bacteria possess probiotic activity. *L. rhamnosus*, also present in NFCBs (Table 1), was efficient in treatment and prevention of gastrointestinal disease [152]. The traditional Romanian NFCB, *borș*, has been consumed since ancient times as a gastric remedy. Several potentially probiotic bacteria—*L. casei*, *L. plantarum*, *L. brevis*, and *L. fermentum*—were isolated from borș, explaining the traditional consumption of this beverage for curative purposes [56].

The predominant microorganisms in the spontaneous fermentation of the African *mahewu*, a non-alcoholic sour beverage made from corn meal and consumed in Africa and some Arabian Gulf countries, belong to *Lactococcus lactis* subsp. *lactis*. On the other hand, the microbiota identification of Bulgarian boza shows that it mainly consists of lactic acid bacteria and yeasts, such as *Lactobacillus plantarum*, *L. acidophilus*, *L. fermentum*, *L. coprophilus*, *Leuconostoc raffinolactis*, *Ln. mesenteroides*, and *Ln. brevis* and *Saccharomyces cerevisiae*, *Candida tropicalis*, *C. glabrata*, *Geotrichum penicillatum*, and *G. candidum*, respectively [19].

In the case of Turkish *boza*, the use of LAB and yeast isolates as starter cultures allows for controlled fermentation studies to be carried out. The selection of proper strains with probiotic and antimicrobial properties enhances the functional properties of *boza* [59].

As seen in the case of African countries, the primary challenge for the development and use of fermented cereal-based probiotic beverages is the common lack of knowledge regarding the health and nutritional benefits of such foods and beverages. Insufficient common knowledge on probiotics and their benefits creates a sense of scepticism among consumers. Moreover, there is also an imperative need to ensure proper facilities for probiotic starter cultures, given that through spontaneous fermentation, the organoleptic and functional qualities of the resulting products are variable [153].

### 2.4. The Nutritional and Bioactive Composition of Commonly Consumed NFCBs

Consumers are aware of the importance of maintaining a strong immune system to prevent illnesses and they are actively looking for products which can help maintain their health status and alleviate health problems. It has been scientifically proven that probiotics isolated from functional beverages boost the immune system [154], and that, along with prebiotics, they are able to improve the intestinal homeostasis, immunomodulating ability, and general health of the host [27].

Looking at past publications, it is shown how fermented beverages have transitioned from traditional natural fermented products to beverages formulated with functional ingredients meant to offer cardiovascular health benefits, and then to functional fermented drinks which improve gastrointestinal health, which could then evolve into fermented products containing specific bioactive nanoparticles [134].

NFCBs are receiving increased attention from researchers and consumers more recently due to their proven probiotic characteristics and disease prevention perspectives [17]. The perceived health outcomes of fermented beverages are strongly related to the microbial content and implicit improvement of gastrointestinal health [38]. Moreover, non-alcoholic fermented beverages offer a sense of wellbeing, as they stimulate the metabolic system [126].

Fermented cereal-based foods, including NFCBs, are a potential source of new functional lactic acid bacteria species besides various nutrients and bioactive compounds, with beneficial effects on human health [56]. Furthermore, as briefly mentioned previously, NFCBs are healthy alternatives to the traditionally consumed food probiotics of dairy origin (e.g., yogurt, kefir, etc.), especially for people with lactose intolerance and milk protein allergies [27].

Given the functional components of fermented beverages and their bioactive compounds released through fermentation by cultures, NFCBs have been linked with many potential and some proven health benefits and actions on digestive, endocrine, cardiovascular, immune, and nervous levels [155]. They present beneficial actions for vital body functions and contribute to the prevention and reduction of risk factors for various diseases [19,23]. NFCBs are rich sources of minerals, vitamins, fibre, flavonoids, phenolic compounds, antioxidants, omega-3 fatty acids, plant extracts, sterols/stanols, amino acids, and biopeptides, among others, which could also protect from oxidative stress and inflammation diseases [17,134]. The presence of numerous valuable compounds in NFCBs grants several health benefits upon consumption, as presented in Figure 3.

Considering group B vitamins, these are unequally distributed in grain tissues. What counts the most in terms of their potential functionality in cereal-based food and beverages is their biological availability. During thermal food processing, these vitamins are destroyed almost completely, however, lactic acid fermentation represents a great tool for food industrialists interested in developing novel vitamin-fortified products. For example, Capozzi et al. (2012) pointed out the ability of many lactic acid bacteria strains, such as *Lactobacillus delbrueckii, Lactobacillus plantarum, Lactobacillus rhamnosus, Lactobacillus fermentum*, and *Pediococcus lactis*, reported in the composition of several NFCBs (Table 1), to biosynthesize riboflavin [156].

A new concept attributed to non-alcoholic cereal-based beverages is that they are healthy drinks with important impacts on human health. It has been shown that consuming NFCBs leads to liver function improvement, increased levels of lactobacilli and bifidobacteria in the intestinal microbiota [157], balanced gut microbiota, and the prevention of bacterial translocation with reduced incidences of nosocomial infections [27]. Furthermore, because non-alcoholic fermented beverages contain several LAB metabolites, their consumption confers bactericidal, bacteriolytic, and bacteriostatic properties, resulting in therapeutic effects at a digestive level. Antimicrobial compounds identified in functional beverages exhibited activities against several Gram-positive and Gram-negative bacteria and against yeasts and moulds [17]. The functional compounds and potential or proven health benefits of NFCBs are presented in Table 1.

As previously discussed, the potential benefits of NFCBs can be ensured in a safe and non-contaminated production environment to prevent the risk of product spoilage. Moreover, there are diverse nutritional and bioactive components of NFCBs, depending on the cereal substrate, microorganisms, and fermentation parameters, and on the technological process followed.

Currently, novel NFCBs are designed based on traditional recipes for delivering improved functional fermented beverages. Such an example is “*amahewu*” or “*mahewu*”, a southern African LAB-fermented non-alcoholic maize-based beverage, which is deficient in vitamin A. In a study published in 2015, the traditional drink was redesigned using provitamin A-biofortified maize in exchange for the traditionally used white maize and it resulted in a functional beverage addressing vitamin A deficiency, a concerning health problem in sub-Saharan Africa [43].

#### 2.4.1. Phenolic Compounds and Antioxidant Activity

Cereal grains and germs contain various major phytochemicals, such as phenolic acids, flavones, phytic acid, flavonoids, coumarins, and terpenes. The bran layers of cereal grains are relatively rich in water soluble and liposoluble antioxidants. Oats include phenolic acids, flavonoids, tocopherols, tocotrienols, and avenanthramides. On the other hand, insoluble fibre of wheat bran contains about 0.5%–1% phenolics and most phenolic compounds, like ferulic acid, are abundant in whole grains [158].

Depending on the cereal substrate, fermentation type, and other added ingredients, each NFCB can have quite different compositions. For example, the phenolic profile of brown beer vinegar indicates that such a fermented beverage is a rich source of phenolic compounds (661 mg GAE/L) and antioxidants, with 30% increased antioxidant activity after acetic fermentation. The same trend was observed regarding some of the drink’s individual phenolic compounds. With cohumulone I (4.44 mg CE/L), cohumulone II (6.58 mg/L), 8-prenylnaringenin (2.33 mg CE/L), 6- prenylnaringenin (1.86 CE/L), humulone (5.62 CE/L), and isohumulone (4.14 CE/L) [159], this non-alcoholic fermented beverage can be considered an alternative source of valuable compounds, which could also be part of specific diets.

Fermentation parameters are of high importance in what concerns an NFCB’s antioxidant profile. For example, a fermentation temperature of 24° C was considered a positive factor influencing the antioxidant activity of *borș*, a Romanian traditional wheat bran-based fermented beverage [55]. The antioxidant activity ranged from 3.05 to 8.52 mmol/L Trolox.

Some of the ancient traditional fermented beverages are currently produced at industrial scale. Still, the original recipes and the nutritional components are being altered because of thermal treatments applied to obtain product safety and stability. The addition of natural starters allowed for the increase in phenolic contents and the enhancement of final antioxidant activity. The same trend considering phenolic content between fermentation stages was found in the case of *borș* [56]. Depending on the processing operations applied, the phenolic compounds ranged considerably (4-hydroxybenzoic acid: 0.9–5.9 mg/L; vanillic acid: 0.6–3.2 mg/L; syringic acid: 0.5–2.5 mg/L; p–coumaric acid: 0.5–1.5 mg/L; sinapic acid: 0.6–2.7 mg/L; ferulic acid: 13.7–47.8 mg/L).

#### 2.4.2. Amino Acids

Although cereals pose few challenges from a nutritional standpoint, especially with regard to the increased starch content upon cooking, the limited amino acid contents of their protein fractions, or the reduced bioavailability of their zinc and iron contents, there are already some solutions meant to correct and further enhance the nutritional status of cereal-based foods and beverages [160].

For example, in a study conducted on *borș*, an increase in amino acid content was found from one fermentation stage to another. The same study stated that the LAB fermentation of cereals improves the protein quality as well as the level of certain free amino acids by enhanced endogenous proteolysis and/or microbial action. Among the identified amino acids, there were isoleucine and threonine [56].

In general, the natural fermentation of cereals allows for amino acids to be synthesised, while it also helps to enhance the availability of group B vitamins. Fermentation processes also offer suitable pH conditions for phytate enzymatic degradation. Such a reduction in phytate may increase the amount of soluble iron, zinc, and calcium several fold, correcting the nutritional status of cereal-based beverages and foods [19].

## 3. Future Perspectives to Enhance the Functional Properties of NFCBs

### 3.1. Consumers’ Preferences and Requirements

The way we perceive healthiness in foods varies between cultures and is reflected in the consumers’ familiarity with health-related information [161]. Moreover, modern-day buyers might show increased interest in foods which can improve well-being [162], reduce the risk of developing illnesses, and satisfy nutrition-related conditions such as food intolerances and allergies. Currently, consumers approach the concept of wellness with a holistic view and are becoming more health conscious, particularly due to the growing incidence of diseases such as type 2 diabetes, coronary heart disease, cancer, and obesity [163]. There is a heightened demand for food products which are nutritious, functional, attractive, with clean labels, and which are ready to eat. Considering such attributes, the beverage sector seems to be one of the most suitable for addressing customers’ demands and for increasing the rate of functional product consumption. Market data indicate that functional food products are among people’s choices even when economic issues arise. The global functional beverage market has increased rapidly, and it is expected to grow further, due to their sensory appeal and health-promoting attributes [38].

Studies showed that providing information on labels regarding health-related claims associated with functional beverages determines consumers’ preferences for a specific product [164]. Not only health-related claims but also the sensory properties determine consumers’ preferences for a specific functional beverage [34]. It was suggested that the importance of specific testing conditions for functional beverages will help product developers to reformulate the product with the proper design of experiments, focusing on the consumers’ needs [165], as they will determine the product acceptance [166]. The urgent need for educational campaigns designed with accessible and easily understood wording about the nutritional components of foods has already been pointed out [167]. Although the previously mentioned study was conducted in Brazil, insufficient knowledge about nutritional facts for food products is a worldwide issue strongly related to the level of education [168].

### 3.2. Possibilities of Improving the Appeal and Functionality of NFCBs

The functional beverage market is competitive and driven by product innovation and health awareness trends concerning optimum nutritional diets. The biological activities and the sensory properties of a beverage arise from individual components, along with chemical and physical interactions within the food matrix during processing, storage, ingestion, and digestion [169]. Beverages deliver nutrients, bioactive components, antioxidants, vitamins, minerals, fatty acids, plant extracts, probiotics, prebiotics, and micronutrients [13]. Cereal-based beverages include a complex mix of different polymers, such as proteins, polyphenols, and polysaccharides. These polymers affect the sensory perception of beverages in terms of mouthfeel, depending on their substance properties [112].

Several strategies have been set to intensify the production, availability, accessibility, and consumption of beverages rich in bioactive compounds. These include combining cereals with pseudocereals (e.g., quinoa, amaranth, etc.), legumes, vegetables extracts, fruits, aromatic plants, and herbs to improve the quality of the final food product [19,56,170,171]. Furthermore, mixing cereals with legumes could improve protein quality [172]. Fruit juices (e.g., apple, pineapple, mango, orange, lemon, peach, lychee, and strawberry) are considered as health-promoting foods and are an important basis of enrichment on which to append an extra functional constituent that can significantly augment the appeal to customers. Cereal beverages are based on grain suspensions. The viscosity, mouthfeel, and sweetness of the drink can be adjusted to the consumers’ tastes using enzyme compounds. Wort can be mixed with various plant- and fruit-based juices to obtain a beverage rich in dietary fibre suitable even for athletes and sport amateurs. Moreover, innovative flavours can be obtained by fermenting the extract with specific microorganisms. As previously mentioned, fruits were also employed in the overall production scheme of fermented cereal beverages, as these are the added sugar necessary for initiating the fermentation process. Chemical analyses of ancient organics absorbed into pottery jars from the early Neolithic village of Jiahu, Henan Province, China have revealed that a mixed fermented beverage of rice, honey, and fruit (hawthorn fruit and or grape) was being produced as early as the seventh millennium B.C. [173]. Such findings inspire the development of new functional products, such as yogurt-like beverages made of a mixture of rice, barley, emmer wheat, oats, soy, and grape must [174] or of other milk-based functional drinks using fermented plant juices [175,176].

Plants are valued for their nutrients such as vitamins, dietary fibre, antioxidants, and flavonoids, which have shown nutritious and health-promoting properties [134]. Bioactive compounds derived from fruits and vegetables can be good vehicles for probiotics, prebiotics, and synbiotics (e.g., fermented cereal beverages) [57,58]. Spices such as tarhana herb, mint, and thyme are mixed with wheat flour, yogurt, vegetables, and herbs to produce a traditional Turkish fermented food called tarhana [177,178] and improve taste, aroma, and other profile characteristics. Moreover, it is suggested that adding herbs with proven beneficial compounds to otherwise traditional fermented soft drinks can augment the nutritional profile of fermented cereal beverages and add therapeutic potential [179]. Among the suggested herbs, we can find echinacea (*Echinacea angustifolia*) for antibiotic action and immune system support, ginkgo (*Ginkgo biloba*) for enhancing memory and alertness, guarana (*Paulina cupana*) for improved cognitive performance, kava (*Piper methysticum*) for stress reduction and mental balance, and St John’s Wort (*Hypericum perforatum*) for anxiety reduction [180]. Vegetables such as beets, tomatoes, carrots, and cabbage can also be included in the production of some functional NFCBs, as they provide supplementary fermentable substrates and nutrients, and act as prebiotics in the final product [181]. *Shalgam* is a traditional Turkish NFCB made of turnip bulb, purple carrot, salt, sourdough, bulgur, or bulgur flour [182], and is thought to regulate the digestive system’s pH and to act as an antiseptic agent. The results of a research study conducted on the co-culture probiotic fermentation of a protein-enriched cereal medium suggests that plant protein may be exploited for achieving protein supplementation of NFCBs. Legumes like chickpeas, individually or in combination with cereals, can provide a good substrate for probiotic microorganism production [145]. Additionally, the composition of legumes, vegetables, and fruits is known to be rich in protein, phytochemicals, dietary fibre, vitamins, and other micronutrients beneficial for human health [183].

Currently, concepts like “green consumerism” and “minimally processed foods” are on the rise, as consumers prefer food lacking synthetic additives. Natural compounds that can be added to functional beverages, such as essential oils and plant extracts including rosemary, peppermint, bay, basil, tea tree, celery seed, and fennel with antimicrobial activity, might represent an alternative to synthetic preservatives. The challenge is to define the optimal dosage of such compounds to have a positive impact on the product’s final nutritional status, sensory properties, and consumer acceptance [12].

### 3.3. Perspectives for Future NFCBs

Due to the increasing prevalence of lactose intolerance and milk protein allergies, as well as the general aim of controlling cholesterol intake, and along with the growing interest in vegetarianism, research efforts have been focused on the feasibility of using cereals as fermentation substrates for the development of probiotic, prebiotic, and synbiotic beverages [27,113,184]. In the production of various non-alcoholic drinks, probiotic lactic acid bacteria are used to boost the product’s functional value [143]. Although ensuring nutritive compounds within NFCB is a crucial step for product development, flavour improvement remains one of the main challenges for the development of LAB-fermented beverages obtained from cereal-based substrates [185].

It is a fact that improved and more appealing NFCB sensory properties would help consumers shift their interest intensively towards such healthy products. The efficacy of probiotics in the treatment of bowel disorders, the prevention of antibiotic-associated diarrhoea, and the improvement of lactose metabolism has been proven [186]. It has been also concluded that both fermentation and acidification with lactic acid have the potential to improve the nutritional quality of cereal-based foods as a method to combat protein malnutrition and iron and zinc deficiencies [187].

Biotechnological processes, such as malting and lactic acid fermentation, are recommended for producing functional beverages with increased contents of functional bioactive components [188]. There are numerous important benefits of enhancing NFCBs through biotechnological processes and it is worth mentioning the reduction of phytates through LAB-fermentation, which in turn leads to the increased absorption of Fe and other minerals [189]. Due to cereal fermentation, quantitative and qualitative changes take place in small molecules, ensuring the high bioavailability of macro- and micronutrients [190,191]. As mentioned earlier, prebiotics and dietary fibre can be added to fermented drinks to heighten their nutritive and functional contents. For example, it was shown that the addition of soy fibre not only improves *Lactococcus lactis* counts but also enhances the beverage characteristics regarding acidity, viscosity, and syneresis [192].

Currently, emerging techniques, such as nanotechnology, are in discussion to enhance the nutrient composition and functionality of NFCBs. However, continuous research and technological improvements are required to better design NFCBs and ensure product safety. It is also crucial to improve the quality of the main ingredients and it is imperative to integrate food safety management systems for industrial scale production. The proposed solutions include the development of new production technologies for obtaining functional NFCBs by extending the spectrum of raw materials used and applying new biotechnological resources (enzymes and lactic acid bacteria) [71].

## 4. Conclusions

Science and technology have the potential to produce superior functional NFCBs and to deliver consistent product quality, to improve shelf life, and to enhance nutritional values to finally meet the consumers’ demands. Past research studies have proven that lactic [55] and acetic [159] fermentation enhance the nutrient content and their bioaccessibility in the case of cereal-based fermented beverages. Adding probiotics and prebiotics to a beverage is perhaps one of the most convenient ways of turning it into a functional beverage.

The research efforts on the enrichment of non-alcoholic fermented cereal beverages are still in their early stages but appear to be more promising than ever. Upcoming studies should focus on traditional non-alcoholic fermented beverages around the world to identify new potential probiotic microorganisms and starter cultures, new ingredients as potential substrates, and to elucidate the contributions of microorganisms and enzymes to the fermentation process.

Knowledge about processing applications and bacterial strains is essential to control the production methods and to design proper mixes of microorganisms, ensuring product safety. Afterwards, starter cultures with expected outcomes can be used for the industrial production of standard-quality fermented beverages with functional attributes.

The effect of advanced technologies on NFCB functional properties during processing requires additional studies to ensure that these technologies can prevent the loss of product quality and nutritive compounds. Moreover, for both scientific and industrial actors, the main challenge is to manage the large-scale production of fermented beverages without losing the unique flavours and other properties associated with the original products. Given this, it is highly recommended to explore the sensory properties of NFCBs when obtained from a combination of cereals, legumes, fruits, and plants.

The current review investigates some of the most scientifically documented traditional non-alcoholic beverages from all over the globe, highlighting their functional compounds and associated health benefits upon consumption. Moreover, processing technologies and their outcome were highlighted and discussed from a large-scale production standpoint. NFCBs were also reviewed, concerning fermentation microbiota and product safety, highlighting the need to apply starter cultures to ensure food safety and standard quality. The nutritional and bioactive compositions of commonly consumed NFCBs were reviewed, showing the functional compounds of NFCBs and their associated health benefits.

NFCBs are associated with functional benefits to one’s health status, as discussed. However, new studies should be carried out to produce new NFCBs, combining probiotic fermented beverages and products such as fruit juices, vegetables, and cereals, meant to address specific health concerns. Future cereal-based fermented beverages need to be balanced regarding sensory properties, nutritional composition, alcohol content, and resource investments to be even more attractive, healthy, and affordable.

## Figures and Tables

**Figure 1 foods-09-01031-f001:**
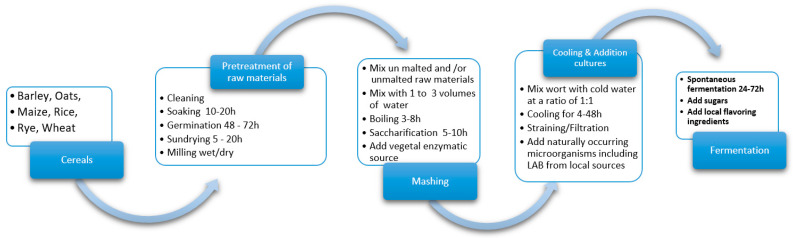
General flow diagram of traditional NFCB production.

**Figure 2 foods-09-01031-f002:**
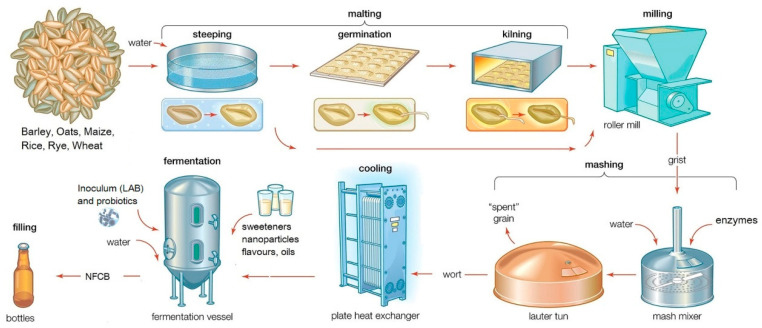
General flow diagram of industrial NFCB production (compilation from Encyclopaedia Britannica’s brewing process).

**Figure 3 foods-09-01031-f003:**
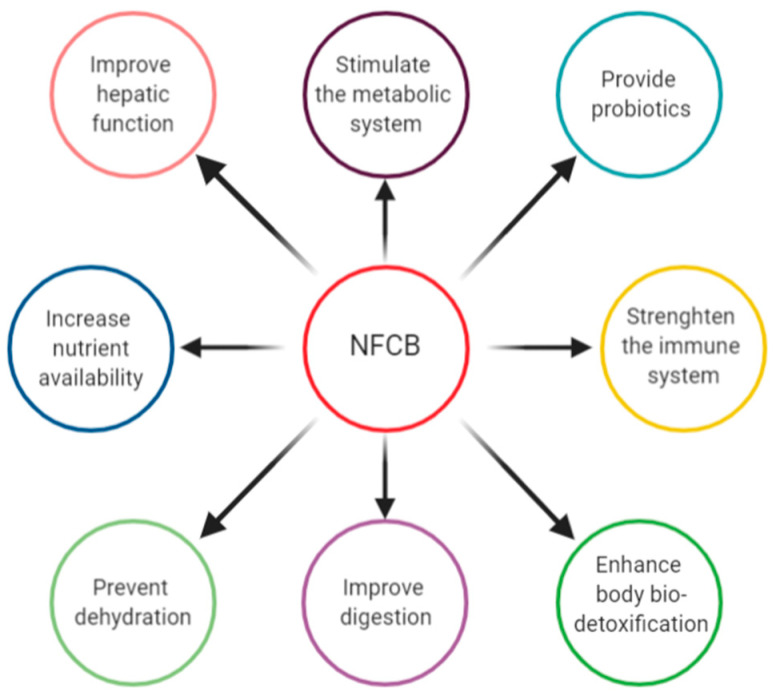
Health benefits associated with NFCB consumption.

**Table 1 foods-09-01031-t001:** Non-alcoholic fermented cereal beverages and their microorganisms, nutritional composition, and health benefits.

Beverage/Place of Origin	Cereals	Microorganisms	Functional Compounds	Health Benefits	Ref.
*Amazake*–Japan	rice *koji*	*Lactobacillus sakei*, *Aspergillus oryzae;*	amino acids; vitamins B1, B2, B6; pantothenic acid, vitamin E, flavonoids, dietary fibre, polysaccharides, sterols;	improves digestion; mitigates hypertension, skin-enhancing action, alleviates liver cirrhosis (200 kcal/150 mL/day/ 12 weeks);	[50,51,52,53]
*Borș*/*Borsht*–Central and Eastern Europe, Romania	wheat bran, corn flour	*Lactobacillus delbrueckii* ssp. *Delbrueckii*;	lipophilic and hydrophilic antioxidants (tocopherols, tocotrienols), phenolic compounds, *group B vitamins*, vitamin E, alkylresorcinols; lignans;	alleviates respiratory and digestive diseases (indigestion, vomiting), effective management of hepatic and bile diseases, potentially beneficial in cancer treatment;	[54,55,56]
*Boza–*Turkey, Greece, Bulgaria, Albania, Romania, Bosnia Herzegovina; South Africa	barley, oats, rye, millet, maize, wheat, rice	*Lactobacillus plantarum*, *Lactobacillus rhamnosus*, *Lactobacillus pentosus*, *Lactobacillus paracasei*, *Lactobacillus fermentum*, *Lactobacillus brevis*, *C. inconspicua*, *C. pararugosa*;	vitamin A, vitamins B1, B2, B6, nicotinamide; Ca, Fe, P, Zn, Na, β-glucan, dietary fibres;	improves gastrointestinal health, stimulates the immune system, decreases cholesterol level;	[38,41,57,58,59,60,61]
*Busa*–Syria, Egypt, Kenya, Turkistan;	rice or millet	*Lactobacillus* sp. *Saccharomyces* spp.	dietary fibre, amino acids, fatty acids, vitamins B1, B2;	lowers cholesterol and reduces risk of cancer and obesity, lowers blood pressure, beneficial in diabetes;	[42,61]
*Gowé*–West Africa, Benin	malted and non-malted sorghum, maize	*Lb. fermentum*, *Weissella confusa, Weissella kimchii, Lactobacillus mucosae, Pediococcus acidilactici, Pediococcus pentosaceus*;	amino acids (glutamic acid and leucine), minerals (Fe, Ca, Zn, P);	certain lactic acid bacteria strains can help in preventing infections by urogenital pathogens; antimicrobial efficacy;	[62,63,64,65,66]
*Kunun-zaki*–Nigeria	wheat and sorghum /millet, wheat, malted rice	*Lb. plantarum, Lb. fermentum, Lactococcus lactis; Saccharomysces cerevisiae;*	minerals (Fe, Ca, Mg, K);	provides micro- and macronutrients, improves nutritional status;	[67,68,69,70]
*Kvass*–Lithuania, Russia, Eastern Poland	extruded rye, malted barley	*Lactobacillus casei, Lb. sakei, P. pentosaceus, S. cerevisiae;*	vitamins B1, B3, B2, B6; dietary fibres, Zn, Cu, maltose, maltotriose, glucose, fructose;	modulates metabolism, reduces flatulence, alleviates hyperacidity;	[17,71,72]
*Mahewu*/*Amahewu*–Africa (Botswana, South Africa, Zimbabwe)	maize, sorghum, millet malt or wheat flour	*Lb. brevis, L. casei, L. lactis, Lb. plantarum, S. cerevisiae, S. pombe;*	Na, K, Ca, Fe, Zn, Mn; dietary fibre, carbohydrates, *group B vitamins***;**	bacteriostatic and bactericidal properties against enteric pathogens;	[73,74,75,76]
*Munkoyo*–Zambia, Democratic Republic of Congo	maize	*Lb. plantarum* *,Weissella confusa, L. lactis Enterococcus italicus;*	fibre, vitamins B1, B2, B3, B6, B12, Ca, Fe, Zn, proteins, crude fat;	suppresses diarrhoea; anti-allergen, antimicrobial properties;	[22,77,78,79]
*Obushera*– Uganda	sorghum flour or millet, maize	*L. Lactis*, *Lb. plantarum*, *Lb. fermentum*, *Lb. delbrueckii*, *Weissella confusa*;	proteins, minerals, fibre;	NA	[80]
*Oshikundu*/*Ontaku*–Namibia	pearl millet meal, sorghum, or pearl millet malt	*Lb. plantarum, L. lactis, Lb. delbrueckii ssp. delbrueckii, Lb. fermentum, Lb. pentosus, Lactobacillus curvatus;*	shikimic acid, maleic acid, phytic acid, succinic acid; vitamins B1, B2; Ca, Cu, Fe, K, Mg, Mn, Na, S, Zn, P;	NA	[81,82,83]
*Pozol*–South Eastern Mexico, Central America	maize	*Lb. plantarum, Lb. fermentum, Lb. casei, Lb. delbrueckii, Leuconostoc sp., Bifidobacterium sp., Streptococcus sp., Saccharomyces sp.;*	group B vitamins; dietary fibre;	reduces cholesterol levels; improves gastrointestinal health; bactericidal, bacteriolytic, bacteriostatic activities;	[38,84,85]
*Shalgam*–Turkey	bulgur flour (wheat)	*Lb. plantarum, Lb. paracasei, Lb. brevis Lb. fermentum, S. cerevisiae;*	β-carotene, group B vitamins, Ca, Na, Fe;	antiseptic agent; probiotic food; regulates the digestive system’s pH; diuretic action;	[59,86]
*Tobwa* (without malt, only LAB)/*Togwa*–East Africa, Tanzania, Zimbabwe	maize, finger millet (togwa)	*Lb. plantarum, Lb. brevis, Lb. fermentum, Lactobacillus cellobiosus, P. pentosaceus, W. confusa;*	amino acids; dietary fibre;; vitamin B2, B9, B12;	antimicrobial activity; enteropathogenic inhibition of *Campylobacter jejuni* and *Escherichia coli*; eases diarrhoea and prevents malnutrition;	[19,87,88,89]

**Table 2 foods-09-01031-t002:** Non-alcoholic fermented cereal beverages and their sensory properties, nature of use, and fermentation status.

Beverage	Sensory Properties/pH	Nature of Use	The Status of Fermentation/Production Scale	Ref.
*Amazake*	cloudy appearance, sour-sweet taste; pH ~3.9;	dessert, snack, natural sweetening agent, baby food, salad dressing;	homemade and industrialised;	[90,91,93]
*Borș*	sour-bitter taste; odour notes: “bran”, “yogurt”, “goat milk-cheese”, “pungent/sour”, “ripe/fermented fruit”; pH 3.3–4.2;	used as a souring ingredient in soups, nutritious drink;	homemade and industrialised;	[54,55,56]
*Boza*	thick liquid, pale yellow colouring; sweet-sour taste; pH 2.93–3.72;	nutritious food, snack;	homemade and industrialised:	[58,59,60,61,94]
*Busa*	thick homogeneous suspension, light to dark beige; sweet-sour taste; pH 3.4–5.3;	traditionally made and served as an alcoholic drink;	homemade;	[42,95]
*Gowé*	brown/white colour; sweet, acidic, cereal taste; soft texture; pH ~3.5–4.7;	thirst-quenching and energy drink; children’s food;	traditional, small-scale processors;	[65,66]
*Kunun-zaki*	low viscosity, creamy appearance; sweet-sour taste; pH ~3.8;	refreshing drink; nutritious beverage;	homemade, local producers;	[67,68]
*Kvass*	slightly cloudy appearance, light-dark brown colour; sweet-sour taste; pH 3.2–4.3;	soft drink;	traditionally homemade; industrialised differently than the traditional approach;	[71,72]
*Mahewu*/*Amahewu*	creamy colour, sour taste; pH ~3.5;	weaning food for infants, consumed in schools, farms, mines, etc.	homemade, commercially produced in African countries;	[45,74,75,76]
*Munkoyo*	slight yellow colour; sweet, mildly sour taste; pH 3.3–4.2;	consumed at household level; energy drink;	homemade;	[22,77,78,79]
*Obushera*	moderately thick composition, pale brown colour; sweet and sour taste; pH < 4.5;	thirst quencher, social drink, energy drink, and weaning food;	homemade; commercially relevant types: Obutoko, Obuteire, Ekitiribita;	[80,96]
*Oshikundu*/*Ontaku*	white colour, milky appearance, sweet taste; pH 3.3–3.7;	a token of welcome and hospitality; consumed at special events and daily social interactions;	homemade; local producers;	[81,82,83]
*Pozol*	yellow-brown colour; sweet-sour; slightly acidic taste; pH 3.8;	food or refreshing beverage, consumed at religious ceremonies and for its curative properties;	homemade in rural and urban areas of southeast Mexico; small- and large-scale producers;	[85,97]
*Shalgam*	red colour; sour taste;	used as a medicine because of its antiseptic agents;	home-scale level; small scale producers;	[59,86]
*Tobwa*/*Togwa*	opaque and brownish colour; sweet, occasionally sour taste; pH~4;	consumed as a popular energy source;	industrially produced in Tanzania;	[87,88]

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
