# Peer review of "Current Functionality and Potential Improvements of Non-Alcoholic Fermented Cereal Beverages"

_foods, 2020, doi:10.3390/foods9081031_

Round 1

Reviewer 1 Report

This  review deals with different aspects of non-alcoholic fermented cereal beverage. The manuscript is in general well written and organized and the references updated, giving a wide overview of this topic. The technological aspects of the beverage production process are clearly exposed. Anyway, there are some points to be addressed and some suggested rearrangement:

Some paragraph are less accurate and I do not think they give interesting information, so I suggest deepening them or eliminating, in particular:

2.4.1 There are few examples of polyphenolic compounds in non-alcoholic fermented cereal beverage

2.4.2 Amino acid: few examples and  the one about quinoa is not appropriate since it is a pseudo cereal

I think that in this form they can be included in the 2.4 paragraph, not as subparagraph

2.4.3 I think this paragraph should be moved in other part of the manuscript, for instance after the fermentation process, as 2.3.1 maybe. Moreover, it does not provide new useful information.

Since it is quite long I suggest to divide Paragraph 3 in some subparagraphs in order to improve the fluency of the manuscript and make clearer the future perspectives

Check and uniform the references, some of them are not in the right template e.g #4, 5, 6 etc

Please avoid  the use of  microflora or flora throughout the manuscript, it is not correct, change in microbiota e.g line295 bacterial flora, line 308 microbial flora etc

Please avoid the use of “in what regards”(line 227, 315,402 etc), change in with regard to, concerning, regarding etc

Minor points

Line 47 become change in “are becoming”

Line 92-93 I would say of “various potentially beneficial microorganism”

Line 97 have change in” has”

Line 98 add “there is the possibility”

Line 142 the adjective relative is not clear, maybe not constant?

Table 1 If a LAB species is found for the first time, you should write Lactobacillus and successively abbreviate in Lb. Uniform vitamins, sometimes are vitamin group B vitamins, then Vitamins B

spp instead of sp after the genera

Boza carbohydrates as functional compounds?

Busa add spp to Saccharomyces

Gowè I think that Weissella kimchii is missing since it is reference  #63.

Oshikundo lactic and acetic acetic should not be included in functional compounds, these two compounds, especially lactic acid, are produced in all the beverages fermented by lactic acid bacteria

Pozol correct del-brueckii

Tobwa W. confusa instead of confusa, bacteria name in italics

 Line 155 I think that low starch availability is not a parameter for comparing cereal beverage to milk products

Line 167 change natural in “spontaneous fermentation”

Line 215-217 reformulate the sentence

Line 223 eliminate o after mixture

Line 368 maybe “used” instead of uses

Line 376 I do not think ref 137 stated what you reported

Line 411 “lactic acid bacteria strain” instead of lactic strain

Line 412 “106 “ instead of 106

Line 448 increased level of lactobacilli and bifidobcteria, specify where

Line 494 remove a before few

Line 518 Lactobacillus in italic

Line 531 “Beverages deliver” instead of are delivering

Line 548 maybe are is missing ? And contents has to be removed

Line 558 add be after can

Line 572 remove the 2nd act as

Line 574 may or can, not both

Line 619 change in “to produce”

Line 634 change require in “requires”

Author Response

Response to Reviewer 1 Comments

We thank the Reviewer for the careful revision and helpful suggestions. All observations have been considered, and the text has been modified accordingly. All modifications are highlighted in yellow in the revised version of the manuscript.

Reviewer 1

Point 1. 2.4.1 There are few examples of polyphenolic compounds in non-alcoholic fermented cereal beverage; 2.4.2 Amino acid: few examples and  the one about quinoa is not appropriate since it is a pseudo cereal; I think that in this form they can be included in the 2.4 paragraph, not as subparagraph

Response 1: Thank you very much for this helpful suggestion, we agreed to include the subparagraphs 2.4.1 and 2.4.2 in the 2.4 paragraph (see lines 522-565). The quinoa example was also removed, along with the reported reference (Lorusso, A.; Coda, R.; Montemurro, M.; Rizzello, C.G. Use of selected lactic acid bacteria and quinoa flour for manufacturing novel yogurt-like beverages. Foods 2018, 7, 1–20, doi:10.3390/foods7040051.), adding instead a couple of the identified essential amino acids of borÈ™ (see lines 559-560).

Point 2. 2.4.3 I think this paragraph should be moved in other part of the manuscript, for instance after the fermentation process, as 2.3.1 maybe. Moreover, it does not provide new useful information.

Response 2: We agree with the reviewer’s comments. The former subparagraph 2.4.3. is now listed as 2.3.1. We also added additional information to highlight the importance of researching probiotics (lines 436-440; 454-460). Supporting references were added.

Point 3. Since it is quite long I suggest to divide Paragraph 3 in some subparagraphs in order to improve the fluency of the manuscript and make clearer the future perspectives

Response 3: We think your comment is accurate, and we have accepted to divide Paragraph 3 in three subparagraphs (see lines 567-684) for laying out the future perspectives of NFCB in a clearer manner.

Point 4. Check and uniform the references, some of them are not in the right template e.g #4, 5, 6 etc

Response 4: The style of references was checked. Some new references were added (they are highlighted in yellow). Each citation was checked in the text against the References and vice-versa.

Point 5. Please avoid  the use of  microflora or flora throughout the manuscript, it is not correct, change in microbiota e.g line295 bacterial flora, line 308 microbial flora etc.

Response 5: We accepted your recommendation and we replaced them with “microbiota”, as you suggested (Lines 298, 396, 400, and 447).

Point 6. Please avoid the use of “in what regards”(line 227, 315,402 etc), change in with regard to, concerning, regarding etc

Response 6: We accepted your suggestion and we replaced “in what regards” with “regarding” and “concerning”, as you suggested (lines 312, 423, 534, and 687).

Point 7. Minor points

Line 47 become change in “are becoming” – paragraph moved as another reviewer requested, the correction is visible at line 573

Line 92-93 I would say of “various potentially beneficial microorganism” – We agree, it is more accurate, correction visible at current line 67

Line 97 have change in” has” – changed “have” in “has” at line 73

Line 98 add “there is the possibility” – corrected, line 74

Line 142 the adjective relative is not clear, maybe not constant? – changed “relative” with “variable”, line 128

Table 1 If a LAB species is found for the first time, you should write Lactobacillus and successively abbreviate in Lb. Uniform vitamins, sometimes are vitamin group B vitamins, then Vitamins B spp instead of sp after the genera – changes were done in Table 1 (line 138) as seen in yellow highlighter

Boza carbohydrates as functional compounds? – we decided to replace “Functional compounds” with “Nutritional compounds” for more clarity. However, carbohydrates have, in a way, a functional role given that they provide energy and prevent the usage of other energy sources such as lean body mass, adipose tissue (Kokkinidou, S., Peterson, D., Bloch, T., & Bronston, A. (2018). The Important Role of Carbohydrates in the Flavor, Function, and Formulation of Oral Nutritional Supplements. Nutrients, 10(6), 742. doi:10.3390/nu10060742).

Busa add spp to Saccharomyces – correction done in Table 1 on Busa’s line

Gowè I think that Weissella kimchii is missing since it is reference  #63. – correction done, Weissella kimchii was added in Table 1

Oshikundo lactic and acetic acetic should not be included in functional compounds, these two compounds, especially lactic acid, are produced in all the beverages fermented by lactic acid bacteria – we agree with your observation, lactic acid and acetic acid were removed from nutritional compounds as the information was redundant

Pozol correct del-brueckii – it is now corrected, the “-” was removed

Tobwa W. confusa instead of confusa, bacteria name in italics – the bacteria names are now in italic and W. confusa was corrected

Response 7: We agree with your suggestions and we have made changes throughout the manuscript.

Point 8: Line 155 I think that low starch availability is not a parameter for comparing cereal beverage to milk products

Response 8: Thank you very much for your observation! We removed this parameter (line 159).

Point 9: Minor points

Line 167 change natural in “spontaneous fermentation” – changed, line 170

Line 215-217 reformulate the sentence – the sentence was slightly reformulated as seen at lines 206-208

Line 223 eliminate o after mixture – corrected at line 214

Line 368 maybe “used” instead of uses – corrected at line 388

Response 9: We agree with your suggestion and we have made changes throughout the manuscript. Moreover,  we have reformulated the indicated sentence (see lines 206-208).

Point 10: Line 376 I do not think ref 137 stated what you reported

Response 10: Thank you for your comment. The text in discussion is currently seen at lines 371-373. Our intention was to advise that further optimization can be applied to traditional cereal fermented beverages to address current needs and points of interest, such as the need of prolonging shelf life or the possibility of addressing such traditionally-rooted beverages to consumers interested in healthy foods using, for example, oats as a fermentable substrate. Angelov et al. 2018 stated the following: “Based on the increased knowledge on oats high nutritional value and consumers’ demand for healthy foods, several oat-based probiotic beverages were introduced in Europe.”

Point 11: Minor points

Line 411 “lactic acid bacteria strain” instead of lactic strain – done, line 498

Line 412 “106 “ instead of 106 – corrected, line 432

Line 448 increased level of lactobacilli and bifidobcteria, specify where – done, line 503 and added corresponding reference

Line 494 remove a before few - done

Line 518 Lactobacillus in italic - done

Line 531 “Beverages deliver” instead of are delivering – corrected it, line 599

Line 548 maybe are is missing ? And contents has to be removed – added “are” removed “contents” (lines 616-617)

Line 558 add be after can - done

Line 572 remove the 2nd act as - done

Line 574 may or can, not both – corrected, line 641

Line 619 change in “to produce” – changed, line 628

Line 634 change require in “requires” – done, line 702

Response 11: We accepted your suggestions and we have made changes throughout the manuscript regarding small grammatical and textual changes.

Reviewer 2 Report

Review - Current Functionality and Potential Improvements of Non-Alcoholic Fermented Cereal Beverages

This nicely written review highlights processing of cereals to non-alcoholic fermented beverages and their functional properties. The text is well-written and the review is insightful and informative.

I have a few suggestions that to my mind need to be implemented to further improve the paper.

  1. I would suggest to greatly shorten the introduction. When starting to read, I am led to think that the review is on consumer perception towards functional foods. This is not what the review actually is about. The authors could place some of this consumer perspective in the future perspectives section. I would delete the first part almost entirely, especially the section up to line 76. Section 2.1 actually more reads like the introduction to the review.

  1. Table 1. The Table shows functional compounds for all products except for Munkoyo. A recent paper came out showing that this product has increased levels of B-vitamins in comparison to the raw materials. https://doi.org/10.3390/nu12061628

This relates to a more general point regarding the nutritional composition of fermented foods. What nutrients are added through fermentation? Lactic acid bacteria excrete B-vitamins and other functional compounds. Thus the fermented products have higher levels of these compounds than the unfermented raw materials. The authors elude to this (line 177), yet this point could be stressed more throughout the paper – for instance in the section 2.4 and in Figure 3.

  1. Discussion and reflection on the microbial composition come up at various places in the paper (such as line 182...), and there is a special section on this (section 2.3). I actually liked the paragraph starting in line 182, since this highlights the aspect of microbial ecology dynamics. This is core to traditional fermentation. The authors may move this to section 2.3 and expand on it. For instance recent work showed that processing practice variation affects microbial composition of the fermenting microbial community (https://doi.org/10.1016/j.lwt.2020.109451).

  1. In line 314 the authors remark that spontaneous fermentation leads to unsafe products. I feel this statement should be nuanced. Firstly, fermented foods are safer than corresponding unfermented raw materials. Secondly, the widespread use of these foods overall without major microbial health problems shows that generally these foods are safe. Finally, fermentation with complex microbial communities provide higher levels of intrinsic stability to the microbial community due to stabilizing interactions between species that prevent and inhibit the proliferation of invader strains, including pathogenic strains.

https://www.nature.com/articles/ismej2013108?report=reader

https://www.nature.com/articles/s41467-018-05308-z/

This I think means that spontaneously fermented products may have a higher intrinsic microbiological safety than those fermented using a defined starter with only few strains. I do agree that when using a defined starter, the fermentation will be more predictable/reproducible. Although artisanal practice of many fermented products (such as raw milk cheese) has also shown than when standardized processing is applied (and this could be spontaneous fermentation like with raw milk cheese), highly reproducible outcomes can be achieved.

  1. In the section on what makes NFCB safe, I think the main point to mention is that these foods have a low pH, preventing growth of most pathogenic strains. (Theron, Maria M., and JF Rykers Lues. Organic acids and food preservation. CRC press, 2010) This could be worked into the section 2.3. You could highlight here what the common pH range is of NFCB and mention the pH cut-off points for growth of some common food-borne pathogens.

  1. In the section on probiotics, you may highlight a bit more by what mechanism fermented foods can have probiotic properties.

https://doi.org/10.1016/j.tibtech.2012.09.002

Author Response

Response to Reviewer 2 Comments

We thank the Reviewer for careful revision and for helpful suggestions. All observations have been considered, and the text has been modified accordingly. All modifications are highlighted in yellow in the revised version of the manuscript.

Reviewer 2

Point 1. I would suggest to greatly shorten the introduction. When starting to read, I am led to think that the review is on consumer perception towards functional foods. This is not what the review actually is about. The authors could place some of this consumer perspective in the future perspectives section. I would delete the first part almost entirely, especially the section up to line 76. Section 2.1 actually more reads like the introduction to the review.

Response 1: . We agree with your suggestion, the introduction is now shorter, as we have placed consumer preferences in section 3 (lines 567-591).

Point 2. Table 1. The Table shows functional compounds for all products except for Munkoyo. A recent paper came out showing that this product has increased levels of B-vitamins in comparison to the raw materials. https://doi.org/10.3390/nu12061628 This relates to a more general point regarding the nutritional composition of fermented foods. What nutrients are added through fermentation? Lactic acid bacteria excrete B-vitamins and other functional compounds. Thus the fermented products have higher levels of these compounds than the unfermented raw materials. The authors elude to this (line 177), yet this point could be stressed more throughout the paper – for instance in the section 2.4 and in Figure 3.

Response 2: Thank you for your comment. B-vitamins were added in Table 1 for Munkoyo, along with the mentioned reference. Moreover, a new paragraph was added at 2.4. for highlighting the LAB’s influence on the final functional compounds such as vitamins (lines 493-500).

Point 3. Discussion and reflection on the microbial composition come up at various places in the paper (such as line 182...), and there is a special section on this (section 2.3). I actually liked the paragraph starting in line 182, since this highlights the aspect of microbial ecology dynamics. This is core to traditional fermentation. The authors may move this to section 2.3 and expand on it. For instance recent work showed that processing practice variation affects microbial composition of the fermenting microbial community (https://doi.org/10.1016/j.lwt.2020.109451).

Response 3: We agree with the reviewer’s comments. The paragraph you mentioned was moved at lines 329-337. Moreover, we added further lines to highlight the microbial ecology dynamics (lines 338-351). The suggested reference was added.

Point 4. In line 314 the authors remark that spontaneous fermentation leads to unsafe products. I feel this statement should be nuanced. Firstly, fermented foods are safer than corresponding unfermented raw materials. Secondly, the widespread use of these foods overall without major microbial health problems shows that generally these foods are safe. Finally, fermentation with complex microbial communities provide higher levels of intrinsic stability to the microbial community due to stabilizing interactions between species that prevent and inhibit the proliferation of invader strains, including pathogenic strains.

https://www.nature.com/articles/ismej2013108?report=reader

https://www.nature.com/articles/s41467-018-05308-z/

This I think means that spontaneously fermented products may have a higher intrinsic microbiological safety than those fermented using a defined starter with only few strains. I do agree that when using a defined starter, the fermentation will be more predictable/reproducible. Although artisanal practice of many fermented products (such as raw milk cheese) has also shown than when standardized processing is applied (and this could be spontaneous fermentation like with raw milk cheese), highly reproducible outcomes can be achieved.

Response 4: Thank you very much for your observation! We made changes and added further lines to highlight the spontaneous fermentation impact on products (see lines 296-305) at 2.2.4. The suggested references were also added to the manuscript.

Point 5. In the section on what makes NFCB safe, I think the main point to mention is that these foods have a low pH, preventing growth of most pathogenic strains. (Theron, Maria M., and JF Rykers Lues. Organic acids and food preservation. CRC press, 2010) This could be worked into the section 2.3. You could highlight here what the common pH range is of NFCB and mention the pH cut-off points for growth of some common food-borne pathogens.

Response 5: We accepted the suggestion and we have made changes in the section 2.3 regarding the effect of NFCB pH on the growth of food-borne pathogens (see lines 346-351). The suggested reference was added to the manuscript.

Point 6. In the section on probiotics, you may highlight a bit more by what mechanism fermented foods can have probiotic properties. https://doi.org/10.1016/j.tibtech.2012.09.002

Response 6: Thank you for your feedback! Based on your suggestion we explained the conditions needed to be accomplished for a food to have probiotic properties (see lines 436-440). The suggested reference was added.

Reviewer 3 Report

The manuscript is well written and the material well organized and presented. The topic exhibits originallity, nevertheless, some additions should be done according my opinion.

For example, the sensorial properties and nature of each NFCB should be refered in the tables. The nature of use or even the country of consumption, the status of fermentation (homemade or industrialised) could be seperate columns in a table. Other metabolites, like lactic acid, vitamins, biogenic amines or minerals if they exist should be mentioned.

Finally, the commercialisation of indigenous NFCB and examples of commerci ally available functional NFCB should be commented in a seperate paragraph or table, if possible.

Author Response

Response to Reviewers’ comments

We thank reviewers for careful revision. We accepted all the suggestions and modified accordingly the text. All modifications have been highlighted in yellow for making easier their check in the revised version of the manuscript.

Response to Reviewer 3 Comments

Reviewer

Point 1. For example, the sensorial properties and nature of each NFCB should be refered in the tables. The nature of use or even the country of consumption, the status of fermentation (homemade or industrialised) could be seperate columns in a table. Other metabolites, like lactic acid, vitamins, biogenic amines or minerals if they exist should be mentioned

Response 1: We think your comment is accurate, and we have accepted to introduce more information regarding sensory properties / pH, nature of use, the status of fermentation and production scale of NFCB in Table 2. Moreover, the nutritional composition of NFCB were revealed in Table 1.

Point 2. Finally, the commercialisation of indigenous NFCB and examples of commercially available functional NFCB should be commented in a seperate paragraph or table, if possible.

Response 2: Thank you very much for this helpful suggestion, we agreed to include further information regarding the production status and commercialisation of indigenous NFCB (see lines 140-152).

Round 2

Reviewer 1 Report

The manuscript was improved according to the suggestion.

Minor points

Table 1

I think that functional is more appropriate if you list phytochemicals, vitamins, aminoacids etc.   Carbohydrates are present in each beverage in different amount, so they confer nutritional value to all of them. Moreover in Kvass glucose, fructose etc are specified and they belong to carbohydrates class. The suggested reference “Important Role of Carbohydrates in the Flavor, Function, and Formulation of Oral Nutritional Supplements ” is very interesting but they are talking about patients who are malnourished or at-risk for malnutrition, so of course carbohydrates are of great importance for them providing energy etc.

I suggest to replace functional and remove carbohydrates.

Author Response

Response to Reviewer 1 Comments

We thank the Reviewer for their suggestions. All observations have been considered, and the text has been modified accordingly. The modifications are highlighted in yellow in the revised version of the manuscript.

Reviewer’s comments: “Table 1

I think that functional is more appropriate if you list phytochemicals, vitamins, aminoacids etc.   Carbohydrates are present in each beverage in different amount, so they confer nutritional value to all of them. Moreover in Kvass glucose, fructose etc are specified and they belong to carbohydrates class. The suggested reference “Important Role of Carbohydrates in the Flavor, Function, and Formulation of Oral Nutritional Supplements ” is very interesting but they are talking about patients who are malnourished or at-risk for malnutrition, so of course carbohydrates are of great importance for them providing energy etc.

I suggest to replace functional and remove carbohydrates.”

Response: Thank you very much for your comments and suggestions. We agreed to remove carbohydrates from Table 1 as they are intrinsic to NFCB. Moreover, we also agreed to have “Nutritional compounds” changed back to “Functional compounds”, as you suggested.

Reviewer 2 Report

The authors present an interesting review paper. I do not have any further comments. 

Author Response

Response to Reviewer 2

Reviewer 2:

“The authors present an interesting review paper. I do not have any further comments.”

Response: We thank the Reviewer for their positive review and kind comment.
